# Evaluation of a complex intervention (Engager) for prisoners with common mental health problems, near to and after release: study protocol for a randomised controlled trial

Tim Kirkpatrick,[1] Charlotte Lennox,[2] Rod Taylor,[3] Rob Anderson,[3] Michael Maguire,[4] Mark Haddad,[5] Susan Michie,[6] Christabel Owens,[3] Graham Durcan,[7] Alex Stirzaker,[8] William Henley,[3] Caroline Stevenson,[2] Lauren Carroll,[1] Cath Quinn,[1] Sarah Louise Brand,[9] Tirril Harris,[10] Amy Stewart,[1] Roxanne Todd,[2] Sarah Rybczynska-Bunt,[1] Rebecca Greer,[1] Mark Pearson,[3] Jenny Shaw,[2] Richard Byng[1]

For numbered affiliations see end of article.

**Correspondence to**
Dr Tim Kirkpatrick;
tim.kirkpatrick@plymouth.ac.uk

## ABSTRACT

**Introduction** The 'Engager' programme is a 'through-the-gate' intervention designed to support prisoners with common mental health problems as they transition from prison back into the community. The trial will evaluate the clinical and cost-effectiveness of the Engager intervention.

**Methods and analysis** The study is a parallel two-group randomised controlled trial with 1:1 individual allocation to either: (a) the Engager intervention plus standard care (intervention group) or (b) standard care alone (control group) across two investigation centres (South West and North West of England). Two hundred and eighty prisoners meeting eligibility criteria will take part. Engager is a person-centred complex intervention delivered by practitioners and aimed at addressing offenders' mental health and social care needs. It comprises one-to-one support for participants prior to release from prison and for up to 20 weeks postrelease. The primary outcome is change in psychological distress measured by the Clinical Outcomes in Routine Evaluation-Outcome Measure at 6 months postrelease. Secondary outcomes include: assessment of subjective met/unmet need, drug and alcohol use, health-related quality of life and well-being-related quality of life measured at 3, 6 and 12 months postrelease; change in objective social domains, drug and alcohol dependence, service utilisation and perceived helpfulness of services and change in psychological constructs related to desistence at 6 and 12 months postrelease; and recidivism at 12 months postrelease. A process evaluation will assess fidelity of intervention delivery, test hypothesised mechanisms of action and look for unintended consequences. An economic evaluation will estimate the cost-effectiveness.

**Ethics and dissemination** This study has been approved by the Wales Research Ethics Committee 3 (ref: 15/WA/0314) and the National Offender Management Service (ref: 2015–283). Findings will be disseminated to commissioners, clinicians and service users via papers and presentations.

## Strengths and limitations of this study

▶ The study will be a two-centre, randomised controlled trial of a through-the-gate intervention for prisoners with common mental health problems; it will provide much-needed evidence of what works for this difficult-to-engage population.

▶ The primary and secondary outcomes have been selected following extensive piloting work and cover a broad range of outcome domains that could be impacted by the complex intervention.

▶ The study adopts a flexible and pragmatic approach to data collection to try to overcome the challenges of following up this population after release from prison.

▶ The lack of blinding of researchers collecting study data is a limitation of the study design.

**Trial registration number** ISRCTN11707331; Pre-results.

## INTRODUCTION

This paper presents the protocol for a randomised control trial (RCT) to test the effectiveness of a complex 'through-the-gate' intervention for prisoners with common mental health problems. RCTs in prison settings are rare,[1] and we are unaware of any that have focused on responses to common mental health problems. This is a surprising omission, given that the point prevalence of mental health problems among prison populations has been reported as between 50% and 90% both in the UK[2–4] and internationally.[5] In England and Wales, an Office of National Statistics survey reported high

rates of personality disorder (64%), neurotic disorders (40%), drug dependency (43%) and hazardous alcohol use (63%) in sentenced prisoners, with higher rates generally being found in remand prisoners.[6] High levels of suicide, suicidal thoughts and self-harming behaviour have also been reported among both prisoners and ex-prisoners,[7] with the risk of suicide for male offenders leaving prison being eight times the national average.[8 9] Our development work indicated high rates of anxiety and depression, with 47% reaching likely caseness for anxiety, post-traumatic stress disorder (PTSD) or depression while in prison, of which 32% were still 'cases' after release. There was also substantial co-morbidity, especially with substance abuse.[10]

In addition to mental health problems, offenders have wide-ranging personal and social problems, including homelessness, unemployment and broken relationships with both partners and children, and they typically live chaotic lives. In our previous cross-sectional study of 200 offenders (100 serving prison sentences and 100 serving community sentences), 37% reported problems with family relationships; the majority of the sample were unemployed or on long-term sickness benefit (65% in prison and 70% in the community sample), and 26% had ongoing legal or criminal justice issues.[11] These results echo previous surveys of prisoners.[12–15] The Surveying Prisoner Crime Reduction study, which is based on regular interviews with a cohort of prisoners in England and Wales before and after release, shows two-thirds reporting unemployed status before going into custody and 37% reporting the need for assistance in finding accommodation on release.[16] These issues tend to be the focus of offenders' own concerns, indicating a crucial need to address them, as well as providing motivation for change. The international literature identifies similar constellations of inter-related personal and social problems among numerous prisoners leaving custody and similar challenges facing resettlement (or 're-entry') services.[17–21]

The cost of failing to address these issues is high. Those serving short-term sentences place a considerable burden on society. Twelve-month proven reoffending rates for short-sentence prisoners are currently close to 60%,[22] and in addition to the distress and inconvenience commonly experienced by their victims, many 'volume' offences have a surprisingly high financial impact. For example, in 2010, the costs of an average domestic burglary were estimated at £3925 and of a less serious wounding at £9790.[23] Therefore, the potential benefits of addressing these issues, to individuals and communities, as well the financial savings, are significant.

There are complex relationships between mental health, substance misuse, social exclusion and criminal behaviour. However, these tend to be studied separately, with interventions designed to address them being developed and delivered in isolation. An underpinning aim of the Engager intervention is to identify and overcome service barriers, particularly between health and criminal justice sectors, and embed multi-agency working within the intervention.

In the UK, prison healthcare is often provided by a number of different National Health Service (NHS), private, or third-sector organisations, each providing separate primary care, mental healthcare, drug and alcohol services. Opiate substitution services are now generally available in prison, and mechanisms for achieving continuity postrelease are improving. Mental health services for those with severe and enduring mental illness have faced considerable challenges[24] but have improved care for those with psychosis, with new evidence now supporting the development of mental health pathways on release.[25]

By contrast, provision of psychological therapy for offenders with common mental health problems is limited in both prison and community settings.[11] The prison environment complicates diagnostic assessment,[4] and for some, fewer stressors in prison may reduce anxiety, making it difficult to identify mental health problems that may arise postrelease. The focus of hard-pressed prison healthcare staff is on immediate concerns, rather than longer-term postrelease planning. Improving Access to Psychological Therapies (IAPT) services in prisons are still in the early stages of development.

Once released into the community, ex-prisoners with common mental health problems are, in theory, provided for by mainstream statutory services including general practice, community mental health teams and IAPT services. In reality, few access these services. For example, in a previous study, we found an average of only 0.96 contacts with mental health services per offender per year for those reporting common mental health problems,[11] suggesting that a lack of care on release is the norm.[26 27]

Despite negligible uptake and high need, no systems worldwide have been identified for actively engaging offenders with common mental health problems while in prison, providing initial treatment and transferring care to community teams. Many ex-prisoners, like others with common mental health problems complicated by co-morbidity, fall between primary care, IAPT and specialist services.[28–31] They are further disadvantaged by their reluctance both to seek help and to accept mental health diagnoses and by lower levels of general practitioner registration.[10 11 26] Services can also be seen as resistant to offenders and are not designed to meet the needs of those with complex and multiple vulnerabilities.[32] This contrasts with well-established services, together with arrangements for transfer of care, for those with opiate misuse.[33 34] In a relatively small proportion of cases, psychological input and/or general support is provided by statutory or third-sector resettlement services, from probation-delivered thinking skills 'booster' programmes for prisoners on licence, through to volunteer or peer-mentoring services,[35 36] although currently resettlement plans typically contain limited reference to health concerns.

In view of these factors, provision of care for common mental health problems should be considered as part of the range of services making up collaborative care and directed towards improving social outcomes and resettlement. The Engager research programme was designed to develop and evaluate a collaborative care intervention for prisoners with common mental health problems, near to and after release from prison, supporting multiple needs rather than focused on specific diagnoses or on a particular therapy. We describe the methods of the Engager trial here.

## Aims and hypothesis

The Engager trial aims to answer the research question: *What is the effectiveness and cost-effectiveness of the Engager intervention plus usual care, compared with usual care alone, in prisoners with common mental health problems, both before release and for between 3 and 5 months following release from prison?* The primary hypothesis is that the participants receiving the Engager intervention plus usual care (the 'intervention group') will have reduced levels of psychological distress as measured by the Clinical Outcomes in Routine Evaluation-Outcome Measure (CORE-OM)[37] at 6 months postrelease from prison (primary outcome) compared with participants receiving usual care alone (the 'control group').

Secondary hypotheses of the trial are that, compared with the control group, the intervention group will have:

► an increase in the number of self-reported met needs in relation to accommodation, education, work/money/benefits, family/friends/company/intimacy, physical and mental health, safety to self and self-care, safety to others and leisure activities;
► improvements in social outcomes (accommodation, education, employment and benefits);
► a decrease in self-reported substance use and level of dependence;
► positive changes in service use across health, criminal justice, social care and third-sector organisations;
► improvements in psychological constructs related to well-being and desistence;
► improvement in health-related quality of life, well-being-related quality of life, subjective experience of care received and perceived helpfulness of services;
► a reduction in levels of proven reoffending.

The trial will estimate the total cost and per prisoner cost of providing the Engager intervention and the cost-effectiveness of the Engager intervention plus usual care versus usual care alone across health, social care and criminal justice sectors.

The trial also includes a parallel process evaluation, designed to: determine the degree to which the core mechanisms of the intervention were delivered; evaluate the extent to which the core mechanisms of the intervention produced the intended outcomes; identify aspects of the intervention and delivery that could be improved; and explore unintended consequences of the intervention.

## METHODS AND ANALYSIS

### Design

The study is a parallel two-group randomised controlled trial (RCT) with 1:1 individual participant allocation to either the Engager intervention plus standard care (intervention group) or standard care alone (control group), with economic evaluation and parallel process evaluation. The trial is registered as ISRCTN11707331 (4 February 2016).

### Setting

The study will be conducted in two investigation centres (South West and North West of England). Participants will be recruited from three prisons, two in the South West and one in the North West of England. Participants will be recruited in equal numbers from each of the two investigation centres, for both the intervention and control groups. Recruitment and baseline interviews will take place in the prisons, with follow-up interviews taking place in a suitable community location or (for those who are back in prison) in prison. Conduct of the trial in each centre will be led by a local principal investigator, supported by a research team, all of whom have received training in Good Clinical Practice (GCP) and the requirements of the study protocol.

### Study population

Potential participants will be men serving a custodial sentence of 2 years or less, who are within 4–20 weeks from release from prison and who are being released to the geographical area of the study. Potential participants will be identified using the Prison National Offender Management Information System. Female prisoners, men on remand and those with a diagnosis of serious mental illness or on the Offender Personality Disorder Pathway will be excluded from the trial. The full list of participant inclusion and exclusion criteria is provided in box.

Individuals will be approached up to 20 weeks prior to release. Initial contact will be made by a member of the research team. They will be given the Participant Information Sheet (online supplementary appendix 1, information sheet for RCT V5 14.03.17), and the researcher will discuss any queries/concerns with them. Researchers will take consent (online supplementary appendix 2, consent form for RCT V5 24.03.2017) from individuals who wish to participate. All individuals providing written informed consent will complete a short screening interview to identify those currently experiencing common mental health problems or who have experienced common mental health problems in the previous 2 years that impacted on their day-to-day functioning and are likely to experience similar problems on release. The screening interview comprises the Patient Health Questionnaire-9 (PHQ-9),[38] the Generalised Anxiety Disorder-7 (GAD-7),[39] the Primary Care PTSD Screen (PC-PTSD)[40] and a bespoke Historical Common Mental Health Problem screen. The PHQ-9, GAD-7 and PC-PTSD are routinely used in IAPT services and are quick and easy-to-administer screening

## Box    Trial entry criteria

### Inclusion criteria
► Men with prison sentences of up to and including 2 years.
► Being released to the geographical area of the study.
► Having between 4 and 20 weeks remaining to serve in prison.
► Willing to engage with treatment services and research procedures.
► Identified using screening instruments as having, or likely to have following release, common mental health problems.

### Exclusion criteria
► Men on remand.
► Women (numbers are smaller, and prisons are remote; resettlement needs are different; research procedures developed are not feasible for this context). Research will be in male prisons only.
► Those with serious and enduring mental disorder and/or on the caseload of the prison in-reach team.
► Those with active suicidal intent requiring management under the safer custody process or prison in-reach team and where the healthcare team managing the prisoner feels it would be detrimental. Once risk levels reduce individuals in this group will be eligible if not excluded for another reason.
► Those with primary personality disorder who are on the caseload of the Offender Personality Disorder Pathway programme.
► Those who present a serious risk of harm to the researchers or intervention practitioners.
► Those unable to provide informed consent.

tools for depression, anxiety and PTSD, respectively. The researcher will read the questions to the participants, using a narrative conversational format developed in our pilot work to facilitate engagement.[41] Individuals will be considered suitable for inclusion in the study if the screening interview indicates that they:

► have a current common mental health problem as indicated by a score of 10 or more on the PHQ-9 or GAD-7, or 3 or more on the PC-PTSD; or
► have experienced a common mental health problem during the past 2 years, which prevented them from functioning normally in everyday tasks and which is likely to be a problem for them again following their release.[i]

If a participant screens in following this assessment, they will continue to the full baseline interview.

Participants will be informed that participation is voluntary and that they are free to withdraw from the study at any time, and it is stressed that withdrawal from the study will not affect their legal rights. They will also be informed that the researcher has a duty to inform prison

---

[i]Participants will be directly asked whether they had periods of 2 weeks or more in the 2 years before coming into prison when they experienced a problem (eg, stress), whether this affected them functioning normally in everyday tasks and whether they think this will be a problem again once they have been released. These three questions were then repeated for problems involving feeling down or depressed, feeling anxious or worrying a lot, having nightmares or horrible thoughts or having panic attacks. If the participants respond yes to all three questions for any of the five types of problems and could provide an example of how it affected their functioning, then they met this inclusion criteria for the study.

staff if they disclose certain information, such as intent to harm self or others.

### Randomisation
Participants will be individually randomised in a 1:1 ratio to receive either the Engager intervention in addition to usual care, or usual care alone. Randomisation will be achieved by means of a web-based system created by Peninsula Clinical Trials Unit (PenCTU). Randomisation numbers will be computer generated and assigned in strict sequence. Randomisation will be stratified to ensure balance between the two treatment arms across the two investigator centres, with each centre having an independent sequence list for an equal number of participants. At the point of randomisation, participants will be assigned the next randomisation number in the sequence.

Confirmation that randomisation has been performed will be communicated in an unblinded fashion to the investigator site staff and to key members of the central research team, via emails automatically generated by the randomisation website. A researcher (usually the same researcher who conducted the baseline interview) will visit the participant in prison to deliver a letter informing the participant of the randomisation outcome. The researcher will go through the letter with the participant, ensuring that they understand their grouping and when they will be seen next and by whom (researcher or practitioner).

### Intervention
The intervention is designed to engage with individuals with common mental health problems who are close to release, developing a pathway of care in preparation for release and resettlement in the community. The intervention will be delivered in prison between 4 and 16 weeks prerelease and for up to 20 weeks postrelease. Providing they are still willing to engage, all participants will receive the intervention for 8 weeks postrelease. However, for those who need further support, the intervention can continue for an additional 12 weeks, although at a lower intensity. This flexible approach to the length of the intervention followed on from our pilot work, which indicated that, while for many participants 2–3 months was sufficient, others required support from the practitioner for a longer period.

Engager is a manualised, person-centred intervention aiming to address mental health needs as well as to support wider issues such as accommodation, education, social relationships and money management. It was developed by bringing together evidence from a realist review,[42] focus groups, case studies and a formative process evaluation. It will be delivered by experienced support workers and supervisor team leaders with experience of therapy. A mentalisation-informed approach underpins all elements of the intervention. Use of existing practitioner skills (eg, those used in coaching, solution-focused therapy, behavioural activation, cognitive–behavioural therapy) is also key to intervention delivery.

At prerelease stage, practitioner and participant will develop a shared understanding of the participant's needs and goals, recognising the links between emotion, thinking, behaviour and social outcomes. A goal attainment plan will be developed and followed, including liaison with relevant agencies and the participant's social networks. Engagement will be maintained throughout the prerelease period; when required, all-day support will be given on release day.

Following release, the practitioner will provide support for the participant to re-enter the community and engage with services. They will continue to work with the participant and any relevant organisations to help them achieve their goals, while encouraging the participant to take responsibility for self-care. The practitioner will also prepare the participant for the end of the intervention, while liaising with relevant community organisations regarding continuity of care.

## Control group
Individuals in the control group will receive care as usual. In prison, they will be able to access primary care, mental health and substance misuse services, as would usually occur. They will also receive support from criminal justice and any other third-sector organisations as standard. Their use of health, criminal justice and third-sector services will be recorded by means of an adapted Client Service Receipt Inventory (CSRI), and medication usage will also be collected from prison medical records and via participant self-report in the community.

## Contamination
There is unlikely to be significant contamination between the intervention and control arms of the study, although it is theoretically possible for: trainers to train practitioners elsewhere, practitioners to pass on skills and working practices to those treating control individuals, intervention materials to influence practice for control individuals and offenders to influence each other. However, the risk of contamination is considered low primarily because there is no alternative funded pathway for delivery of the substantive components of the intervention for those in the control arm. Engager practitioners form a separate team in prison and while other practitioners are informed about the intervention, (1) they are not trained in the detail, (2) they tend not to have contact with our participants who are selected for the study using case finding and (3) they don't have governance arrangements in place to follow individuals into the community. Cluster randomisation to prevent contamination would have been theoretically possible by randomising at a prison level, but practically not feasible because prisons are clustered together in localities, with one for new entrants, so each cluster would have several prisons. Additionally, the prison system can be subject to sudden and significant changes to prison procedures and entrants and it was estimated that a minimum of six clusters would be required

in order to ensure balance, and this would have incurred prohibitive costs.

## Outcome measures
Outcome measure data[ii] will be collected at approximately 1 week prerelease and at 1, 3, 6 and 12 months postrelease from prison (see table 1). The primary outcome point is at 6 months postrelease. The primary outcome measure is change in levels of psychological distress as indicated by the clinical score of the CORE-OM. This is a 34-item scale comprising four domains, namely: subjective well-being; depression and anxiety symptoms; general, social and close relationship functioning; and items concerning risk of harm to self or others. Items are rated against how participants felt over the previous week, on a 5-point Likert Scale, with eight items reverse scored. CORE-OM was chosen as the most appropriate primary outcome measure at a consensus meeting following a period of pilot testing of a range of outcome measures. In particular, the CORE-OM is a reliable and well-validated measure,[37] was regarded as being quick to administer and easy to understand in pilot testing and was considered to reflect the ultimate aim of the intervention. The primary outcome point is similar to that used in many previous prisoner resettlement studies, 3–6 months postrelease being widely regarded as a suitable follow-up period as the aim of resettlement interventions is to help people reintegrate into community life rather than to provide long-term support.[43] The 12-month follow-up will assess whether any benefits brought about by the intervention are maintained over a longer time period.

Secondary outcome measures are as follows:
► self-reported met need across key outcome domains using an adapted version of the Camberwell Assessment of Need-Forensic Version (CAN-FOR),[44] in terms of the number of met needs and using the Met Needs Index as an aggregate measure of met need[45];
► social outcomes (accommodation, education, employment and benefits);
► drug and alcohol use using and adapted version of the Treatment Outcomes Profile (TOP)[45];
► drug and alcohol dependence using the Leeds Dependence Questionnaire (LDQ)[46];
► service use across health, criminal justice, social care and third-sector organisations using an adapted version of the CSRI.[47] Recent evidence suggests that self-reported health service use data is valid in ex-prisoner populations[48];
► perceived helpfulness of services using the adapted version of the CSRI;
► generic health-related quality of life using the EQ-5D-5L questionnaire[49];

---

[ii]The selection of the primary and secondary outcome measures was informed by two consensus exercises and a period of field testing of a range of possible measures to establish the psychometric properties and acceptability of the measures in this population. This work will be presented in a separate article.

**Table 1** Tabulated summary of study schedule

| Timepoint | | Screening $t_0$ | Baseline $t_1$ | Allocation | Prerelease −1 week $t_2$ | Postrelease from prison +1 month $t_3$ | +3 month[3] $t_4$ | +6 month $t_5$ | +12 month $t_6$ |
|---|---|---|---|---|---|---|---|---|---|
| **Enrolment** | | | | | | | | | |
| Eligibility screen | | X | | | | | | | |
| Informed consent | | X | | | | | | | |
| PHQ-9 | | X | | | | | | | |
| GAD-7 | | X | | | | | | | |
| PTSD-Screening Questionnaire | | X | | | | | | | |
| Historical screen for past CMHPs | | X | | | | | | | |
| Allocation* | | | | X | | | | | |
| **Interventions** | | | | | | | | | |
| Intervention group | Engager intervention | | | | ◆━━━━━━━━━━━━━━━━━━━━━━━◆ | | | | |
| | Usual care | ◆━━━━━━━━━━━━━━━━━━━━━━━━━━━━━━━━━━━━━━━━◆ | | | | | | | |
| Control group | Usual care | ◆━━━━━━━━━━━━━━━━━━━━━━━━━━━━━━━━━━━━━━━━◆ | | | | | | | |
| **Assessments** | | | | | | | | | |
| CORE-OM Questionnaire | | X | | | | X | X | X | X |
| CORE-10† | | | | | | X | X | X | X |
| Adapted CAN-FOR | | X | | | | | X | X | X |
| Adapted CSRI (including medication) | | X | | | X | | X | X | X |
| Objective social outcomes (eg, housing) | | X | | | | | | X | X |
| TOP | | X | | | | X | X | X |
| Leeds Dependence Questionnaire | | X | | | | | | X | X |
| EQ-5D-5L Questionnaire | | X | | | | | X | X | X |
| ICECAP-A Questionnaire | | X | | | | | X | X | X |
| IOMI | | X | | | | | | X | X |
| SAPAS | | X | | | | | | | |
| Neurodevelopmental Symptoms Rating Scale | | X | | | | | | | |
| Trauma Questionnaire | | X | | | | | | | |
| Contact Sheet | | X | | | X | X | | | |
| Brief Inspire Questionnaire | | | | | X | | X | X | |
| Police National Computer Offending Data | | X | | | | | | | X |
| **Safety monitoring** | | | | | | | | | |
| Adverse event reporting | | ◆━━━━━━━━━━━━━━━━━━━━━━━━━━━━━━━━━━◆ | | | | | | | |

*Allocation will be performed using a web-based system provided by the clinical trials unit, usually within 2 days of completing the screening interview.
†CORE-10 will only be completed if it is not possible to complete the CORE-OM Questionnaire.
CAN-FOR, Camberwell Assessment of Need-Forensic Version; CMHP, common mental health problems; CORE-OM, Clinical Outcomes in Routine Evaluation-Outcome Measure; CSRI, Client Service Receipt Inventory; GAD-7, Generalised Anxiety Disorder-7; ICECAP-A, ICEpop CAPability measure for adults; IOMI, Intermediate Outcomes Measurement Instrument; PHQ-9, Patient Health Questionnaire-9; PTSD, post-traumatic stress disorder; SAPAS, Standard Assessment of Personality-Abbreviate Scale; TOP, Treatment Outcomes Profile.

► well-being-related quality of life using the ICEpop CAPability measure for adults (ICECAP-A) questionnaire[50];
► experience of care using the Brief Inspire questionnaire[51];
► psychological constructs related to desistence using the Intermediate Outcomes Measurement Instrument (IOMI)[52];
► well-being, functioning, psychological symptoms, and risk using the subscales of the CORE-OM;

► proven reoffending rates, based on data from the Police National Computer.

Due to the nature of the intervention, it will not be possible to blind participants or those delivering the intervention. Attempts to blind researchers during the pilot trial proved challenging and were largely unsuccessful. Therefore, researchers will be aware of which group participants are allocated to, and measures will be implemented to minimise and measure bias, especially for data collection on the primary outcome measure.[41] Specifically, the researchers will use a highly scripted interview for the primary outcome measure, reading each question to the participants and only deviating from this to clarify the meaning of the question if they indicate they do not understand the question.

## Sample size

The sample size is based on the ability to detect a difference on the primary outcome only and not on the inclusion of baseline measures as covariate. On the CORE-OM, 5.0 points is the accepted Reliable Change Index in service evaluations.[53 54] In contrast, 2.5 points is held as the upper limit of what would be considered a change compatible with equivalence (MBarkham, personal communication, 2015) in trials comparing two interventions. Other trials using the CORE-OM for mental health interventions versus treatment as usual or waiting list controls have achieved mean between group differences in change score of between 3.5 and 7.8.[55 56] An SD of 5.6 was found in the pilot trial.[41] However, larger clinical studies have reported larger SDs of approximately 7.5.[57]

Given the uncertainty, in both SD and the appropriate minimally clinically important difference (MCID) for the CORE-OM, we calculated sample sizes for different scenarios based on the range of values for these two parameters (see table 2). This is equivalent to aiming to be able to detect a minimum effect size of 0.26 (ie, small to medium).

Based on the conservative scenario of an MCID of at least 3.5 and a common SD of 7.5, we will require CORE-OM data on 97 participants in each group at 90% power and 5% alpha. Using an attrition rate of 30%, 140 participants are required per group. Follow-up rates of 63% and 55% were achieved in feasibility and pilot work.[41] However,

**Table 2** Sample size (for each group) based on different values of SD and MCID for the Clinical Outcomes in Routine Evaluation-Outcome Measure

| | | SD | | |
|---|---|---|---|---|
| | | 5.5 | 6.5 | 7.5 |
| Change to be detected (MCID) | 5.0 | 26 | 36 | 48 |
| | 4.5 | 32 | 44 | 59 |
| | 4.0 | 40 | 56 | 74 |
| | 3.5 | 52 | 73 | 97 |

At 90% power and two-sided alpha of 5%.
MCID, minimally clinically important difference.

with learning from the pilot trial and assistance from the new Community Rehabilitation Companies (CRCs), which now supervise virtually all prison leavers for at least 1 year, an attrition rate of 30% or less is achievable.

## Trial data collection

Trial data will be collected from participants at baseline, 1 week prerelease and at 1, 3, 6 and 12 months postrelease. Feasibility and pilot studies highlighted that this population often lead chaotic lives and are difficult to follow up in the community.[41] To address this challenge, the research team will make multiple and sustained attempts to follow up participants at each time point. Community follow-up interviews will be conducted in a convenient location for the participants and where appropriate in the premises of services (eg, National Probation Service or CRC) with which the participant is engaged. Participants will be provided with high street shopping vouchers compensating them for their time at the 3, 6 and 12 months postrelease interviews, although this does not apply to participants who have returned to prison and are interviewed there.

The pilot trial highlighted that some participants can be temporarily lost, but subsequently re-emerge (possibly engaging with community services or back in prison).[41] Follow-up data collection points will take place within broad time-windows. The 1-month follow-up will take place between 14 and 60 days postrelease, the 3-month follow-up will take place between 61 and 151 days postrelease, the 6-month follow-up between 152 and 244 days postrelease and the 12-month follow-up between 304 and 483 days postrelease. Where feasible, follow-up interviews will take place as close to 1, 3, 6 and 12 months postrelease time points as possible. Furthermore, if a participant misses a follow-up interview (eg, at 3 months), they will continue to be included in the study until all follow-up time-points have lapsed (eg, 483 days postrelease), after which point those remaining out of contact will be regarded as lost. If the research team is in contact with a participant but setting up a face-to-face interview is challenging (or if they have failed to turn up to an appointment), researchers will attempt to complete the CORE-10[iii] by telephone. However, even when the CORE-10 has been completed, researchers will endeavour to follow up participants with a face-to-face interview.

The numbers and reasons for drop-outs and losses to follow up will be reported for each arm of the study.

### Baseline and 1 week prerelease data collection

Baseline data collection will usually continue immediately after the screening interview, although additional sessions can be arranged to meet the needs of individual participants or time constraints within the prison. As outlined in table 1, the following data will be collected at this point:

---

[iii]The CORE-10 will be used when completing the measure over the phone because it is shorter. The 34-item CORE-OM was considered too long to complete over the phone with this population of participants.

- psychological distress using the CORE-OM;
- subjective rating of need across health and social domains using an adapted version of the CAN-FOR;
- healthcare, criminal justice and other service utilisation using an adapted version of the CSRI;
- objective social outcomes (accommodation, education, employment);
- drug and alcohol use and dependence using the TOP and LDQ;
- health-related quality of life using the EQ-5D-5L;
- well-being-related quality of life using the ICECAP-A;
- IOMI;
- Standardised Assessment of Personality-Abbreviate Scale;
- Neurodevelopmental Symptoms Rating Scale;
- experience of traumatic life events using the Trauma Questionnaire.

The questions from the standardised measures, including the primary outcome measure (CORE-OM), will be read out to participants in a precise and consistent manner, to minimise bias and overcome any literacy problems. Questions from the secondary outcome measures are incorporated into a specially constructed flexible interview, which avoids duplication of subject matter in order to reduce disengagement or irritability. Data will be recorded in the Baseline Case Report Form.

In addition to the baseline data collection, the researcher will complete a contact sheet for each participant. This will include contact numbers and addresses provided by the participant, as well as a list of services that they are likely to be in contact with postrelease. This sheet will be completed in collaboration with the participant, and they will sign the form to consent that the research team can contact them via the relevant services.

The researchers will meet with the participant again within the week prior to their release. The service use table from the adapted CSRI, to collect information on services the participant has seen since the baseline data collection, and the Brief Inspire Questionnaire will be completed to measure the participant's experience of these services. The researcher will also update contact details for the participant.

Information regarding medication prescribed in the 3 months before prison release, as well as any chronic medical conditions or acute conditions in the previous 12 months, will be collected from the prison healthcare records system. Summary data regarding offence history and number of previous custodial sentences will also be collected from prison records.

## Follow-up data collection

At all follow-up meetings, the researcher will remind the participant of the information sheet and consent, drawing attention to data confidentiality and instances of disclosure where the researcher would need to breach confidentiality.

At approximately 1 month postrelease, the researcher will contact the participant. This follow-up can be completed by phone, but preferably face to face to support continued engagement. The researcher will read aloud to the participant and record responses to the CORE-OM. These data will be used in analysis, but the main objective of the meeting is sustained engagement and planning further contact.

The 3-month, 6-month and 12-month follow-ups will take place between 61 and 151 days, 152 and 244 days and 304–483 days postrelease, respectively, although researchers will endeavour to complete data collection close to the 3-month (90 days), 6-month (182 days) and 12-month (365 days) points. At each time point, participants will be read the questions from the measures listed in table 1.

## Economic evaluation

The cost-effectiveness of the intervention to increase engagement and access to services and to improve mental health outcomes will be assessed. This will be compared with service access and support as usual, using the economic model developed in the pilot phase, populated with the trial outcomes and resource use data up to 12 months postrelease from prison. It will be conducted from a public sector perspective, initially with the same time horizon as the RCT, and primarily using a cost-consequence approach. Within the cost-consequence approach, the estimated incremental costs will be compared with:

- The number of people provided with the service/intervention.
- The incremental differences in the main RCT self-reported health outcomes—CORE scores and EQ-5D-5L and ICECAP-A social preference weights.
- Incremental differences in the number of ex-prisoners who: have resettled; are in employment; have no proven re-offending; are not homeless.
- Estimated lifetime gains in quality-adjusted life years (QALYs). Improvements in lifetime gains will be linked with short-term gains seen in the trial and will be associated with social inclusion outcomes such as effective resettlement, increased employment or reduced reoffending rates. Sustained improvements in these will be modelled based on evidence from the literature. We will test the impact of differing the duration of the persistence of any short-term gains on QALYs (and associated costs) if there is no evidence in the literature.
- The cost of providing the intervention will be based on a combination of process of care data collection and intervention practitioner care records and diaries (bottom-up costing approach) and the total costs of service provision (top-down costing). Both deterministic and probabilistic sensitivity analysis will be conducted to explore uncertainty in the model assumptions and parameters, with exploration of key sources of structural uncertainty where feasible.

The analyses will be conducted according to current guidance from the International Society for Pharmacoeconomics and Outcomes Research on best practice for

conducting trial-based economic evaluation.[58] Consistent with the analytical approach used in the statistical analysis of the effectiveness outcomes of the RCT where possible; the cost-effectiveness analysis will be reported in accordance with the Consolidated Health Economic Evaluation Reporting Standards.[59]

### Process evaluation

The process evaluation will be conducted in parallel with the trial and will adopt a mixed-methods, realist-informed, approach.[60] During the development and piloting of the Engager intervention we produced and refined a theoretically informed, and evidence-based, logic model of the ways in which the intervention was understood to work,[61] which we intend to test in the process evaluation. The logic model included the core components of the intervention that the practitioners were asked to deliver, the key mechanisms of impact (ie, how what the practitioners were doing was understood to produce the desired outcomes) and the anticipated outcomes.[62]

### Process evaluation specific objectives

a. to determine the degree to which the core components of the intervention were delivered and the key mechanisms of the intervention occurred;
b. to evaluate the extent to which the core components and key mechanisms of the intervention produced the intended outcomes;
c. To explore any unintended consequences of delivering the intervention;
d. to identify aspects of the intervention and delivery that could be improved;
e. to identify any aspects of intervention delivery that require additional input from practitioner teams when the research team is no longer in place;
f. to develop an understanding of how to deliver the intervention in real-world settings (training, supervision, meta-supervision).

### Data collection

The data collection methods were developed and refined for acceptability in the pilot trial Formative Process Evaluation and include:
► intervention components checklist to measure fidelity to the intervention;
► semistructured interviews, with a purposively selected subsample of participants, some on one occasion and some at regular intervals throughout their participation in the trial;
► semistructured interviews with Engager practitioners and supervisors throughout the trial;
► semistructured interviews with other practitioners and team leaders, in other services about their perceptions of, and interactions with, the Engager practitioners, participants and the intervention;
► semistructured interviews with family/partners/friends of participants receiving the Engager intervention;
► audio-recordings of practitioner group supervision sessions;
► audio-recordings of selected practitioner–participant interactions;
► Engager practitioner records and notes;
► quantitative outcome measures, contained within the case report form (CRF), and also being used as part of the main trial outcomes;
► ethnographic field notes recorded by the process evaluation researchers.

### Data analysis

The framework analysis methodology, which we developed and applied in the Formative Process Evaluation, will be used and extended to collate and interrogate the process evaluation data.[63] The deductive components of the framework will be informed by the logic model's key mechanisms of impact, that is, the ways in which we understand the intervention to be working. Inductive components of the framework will be surfaces as part of the analytical process. At the end of this analytical process, the logic model of the key mechanisms of impact of the intervention will be revised.

The process evaluation researchers will be distinct from the researchers collecting outcome measures. They will contribute to the qualitative, and therefore more subjective, data collection and the overall analysis. A 'critical friend' researcher, external to the outcome measure and delivery teams, will facilitate the process evaluation researchers' opportunity to self-reflexively explore how their presence affects their data collection and experiences in the field, which may influence their analytical processes.[62] When the process evaluation data and analysis can contribute to refining ongoing fidelity to the Engager model, it will be fed back directly to the intervention delivery team. When the process evaluation data and analysis concerns the outcomes of interest, the data will be shared after the trial database has been locked down and initial statistical analyses have been carried out.

If the main trial does not demonstrate that the intervention is effective, additional analysis of the qualitative data will be conducted using thematic methods to explore possible explanations for this[64] and to glean any additional learning that may have application to other studies with socially marginalised populations and/or those with mental health needs.

### Serious adverse events

Non-serious adverse events will not be recorded. Serious adverse events (SAEs) will be recorded and reported. Any SAEs deemed to have a causal relationship to trial participation with be reported to the sponsor within 24 hours of the chief investigator being informed.

### Study timeline

Study start date: 14 January 2016.

Trial registration date: 4 February 2016.[iv]

Projected end date for recruitment: 30 September 2017.

Projected end date for 6-month follow-up data: 31 July 2018.

Projected initial analysis of primary outcome data: 30 November 2018.

Projected final report date: 31 October 2019.

Current status: recruiting.

### Data management and statistical analysis plan

All data will be treated confidentially and stored securely and anonymously. CRFs will be checked and signed at the research sites by a member of the research team before being sent to the PenCTU for double-data entry on to a password-protected database. All forms and data will be tracked using a web-based trial management system. Double-entered data will be compared for discrepancies, and discrepant data will be verified using the original paper data sheets.

All quantitative data analyses will be conducted and reported in accordance with Consolidated Standards of Reporting Trials recommendations. We will closely monitor the process of data collection during the trial providing flow diagrams summarising, by group, the numbers approached, recruited, randomised, followed up/lost to follow-up and outcome completion.

Primary analyses will be conducted on an intention-to-treat basis (ie, according to randomised group) and compare primary and secondary outcomes at 6-month follow-up between randomised groups on those with complete data sets. Outcomes will be compared using linear-regression-based methods, adjusting for baseline outcome scores and stratification variables (eg, investigation centre). Where necessary, outcomes will be transformed to ensure good regression model fit. A secondary analysis will compare primary and secondary outcomes between groups at all follow-up time points using a repeated measures approach. Reasons for missing data (including loss to follow-up and participant drop-out) will be documented and the baseline characteristics of those with and without missing data compared. Using different assumptions for missing data, we will undertake sensitivity analyses using various imputation models, comparing between group results to the completers' primary analysis. We shall also explore the possibility of conducting secondary per protocol between-group comparisons. If possible, this will be based on a predefined minimum level of intervention receivership and using complier average causal effect analysis methods. The analyst will

be blinded to group allocation and the analysis will be undertaken using STATA V.14.2.

A detailed statistical analysis plan will be prepared before any data analysis is conducted. The statistical analysis plan will be agreed with the trial steering committee.

### Trial management and independent committees

Members of the research team directly involved with the day-to-day running of the trial will meet fortnightly to discuss trial progress, with additional email and telephone exchanges as required. A full trial management group including health economists, statisticians, process evaluation researchers and a sponsor representative will meet quarterly to review trial progress.

The Engager trial steering committee (Chair: Professor Pamela Taylor and three other independent members including a patient and public involvement representative) will meet 1–2 times per year to oversee the conduct of the trial, safety and ethics. The trial steering committee formally agreed that given the social/psychological nature of the intervention, only limited safety monitoring would be required, and therefore an independent data monitoring committee was not required.

### Ethics and dissemination

We have obtained National Offender Management Service (ref: 2015–283) approval and local Trust governance approvals for each site (Devon Partnership NHS Trust, Dorset Hospital University Foundation NHS Trust and Lancashire Care NHS Foundation Trust). The study has also been adopted by the National Institute for Health Research (NIHR) Clinical Research Network and the study sponsor is Devon Partnership NHS Trust.

The trial will be conducted in accordance with the ethical principles outlined in the Declaration of Helsinki and those consistent with GCP. The trial steering committee will ensure adherence to these guidelines. Any amendments to the protocol will be submitted for ethical approval as appropriate.

Findings will be published in peer-reviewed journals and presented at local, national and international conferences to publicise and explain the research to key audiences. A final report will be submitted to NIHR.

**Author affiliations**
[1]Plymouth University Peninsula Schools of Medicine and Dentistry, Plymouth University, Plymouth, UK
[2]Division of Psychology and Mental Health, University of Manchester, Manchester, UK
[3]Exeter Medical School, University of Exeter, Exeter, UK
[4]Centre for Criminology, University of South Wales, Pontypridd, Wales
[5]School of Health Sciences, City, University of London, London, UK
[6]UCL Centre for Behaviour Change, University College London, London, UK
[7]Criminal Justice Programme, Centre for Mental Health, London, UK
[8]LIFT Psychology Service, Avon & Wiltshire Mental Health Partnership NHS Trust, Swindon, UK
[9]Y Lab, Cardiff University, Cardiff, Wales, UK
[10]Institute of Psychiatry, Kings College London, London, UK

**Contributors** TK, CS and LC wrote the first draft of the manuscript. RB is the chief investigator on the National Institute for Health Research grant and RST, RA, MM,

---

[iv]The trial was retrospectively registered about 3 weeks after the first participant was recruited. The trial was registered on the National Institute for Health Research Portfolio Database on 15 December 2015, but we encountered delays in the information being transferred for trial registration. We only became aware of this shortly after we had started recruitment and the issue was quickly rectified and the trial was registered on 4 February 2016.

MH, SM, CO, GD, AS, WH, CQ, TH, MP and JS are co-applicants on the grant and contributed to the conceptualisation of the study design. CL, SLB, ASte, RT, SRB and RG are members of the study team that have contributed to the analysis of pilot data and development of the trial methodology. All authors provided critical evaluation of the manuscript and have given final approval of the manuscript.

**Funding** Engager is funded by the National Institute for Health Research (NIHR) under its Programme Grant for Applied Research Programme (grant number: RP-PG-1210-12011). This research was also supported by the NIHR Collaboration for Leadership in Applied Health Research and Care South West Peninsula at the Royal Devon and Exeter NHS Foundation Trust. The views expressed are those of the author(s) and not necessarily those of the National Health Service (NHS), NIHR or the Department of Health. The funder had no role in the design of this study and will not have any role during execution, analysis, interpretation of findings or decision to submit results. The views expressed are those of the authors and not necessarily those of NHS, NIHR or the Department of Health.

**Competing interests** None declared.

**Patient consent** Obtained.

**Ethics approval** NHS Research Ethics Committee (Wales REC 3, reference: 15/WA/0314).

**Provenance and peer review** Not commissioned; externally peer reviewed.

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
