## [Reviewer comments · BMJ Open]

ARTICLE DETAILS

TITLE (PROVISIONAL)	Evaluation of a complex intervention (Engager) for prisoners with common mental health problems, near to and after release – study protocol for a randomised controlled trial.
AUTHORS	Kirkpatrick, Tim; Lennox, Charlotte; Taylor, Rod; Anderson, Rob; Maguire, Michael; Haddad, Mark; Michie, Susan; Owens, Christabel; Durcan, Graham; Stirzaker, Alex; Henley, W; Stevenson, Caroline; Carroll, Lauren; Quinn, Cath; Brand, Sarah; Harris, Tirril; Stewart, Amy; Todd, Roxanne; Rybczynska-Bunt, Sarah; Greer, Rebecca; Pearson, Mark; Shaw, Jenny; Byng, Richard

VERSION 1 – REVIEW

REVIEWER	Stuart Kinner Murdoch Children's Research Institute & University of Melbourne, Australia
REVIEW RETURNED	14-Jul-2017

GENERAL COMMENTS	This paper describes the protocol for what is potentially a significant trial in an area that has suffered from a dearth of rigorous evaluation research. The inclusion of a process evaluation and health economic analyses should be commended. The focus on continuity of care, and on care coordination, is sensible and consistent with the evidence. My two main issues with the current draft of the manuscript are: - The primary outcome is psychological distress. This is the only outcome for which power calculations are provided. Given the scope of the intervention, why did the researchers decide to focus on this outcome in particular?- In my view the manuscript could be shortened considerably, without loss of information. A careful proofread for clarity of expression would also strengthen the manuscript. I would also encourage the authors to consider replacing the term 'offender' with something less pejorative. Since the population of interest is adult males released from prison, not necessarily on parole, one might argue that use of the term 'offender' in this context is somewhere between pejorative and inaccurate. More specific comments follow. P. 5 line 5 "rates of 50-90%" – presumably prevalence rates? Is this point, period or lifetime prevalence? p. 5 line 5 "trials are limited" – it might be appropriate to cite here a recent systematic review documenting precisely this: Kouyoumdjian et al (2015). Am J Public Health, 105(4), e13-e33.
---

	P. 5 line 11 need a reference to support the statement about PTSD. p. 5 line 13 note also recent evidence of high rates of self-harm after release from prison: Borschmann et al (2017). ANZ J Psychiatr, 51(3), 250-259. p. 5 line 16 clarify whether comorbidity refers to multiple mental illnesses, or co-occurrence of mental illness and substance use disorder, or both/either. p. 5 line 21 “in our previous study” – more information is required and could be provided in a few words – for example was this a cross-sectional study of ### prisoners in the UK? p. 5 line 25 – is the SPCR study based in the UK? p. 6 line 15 note that there is also some international evidence regarding engagement with mental health services for ex-prisoners: see for example Thomas et al (2016). Psychological Medicine, 46(3), 611-621. p. 6 line 49 it would be helpful to briefly explain WHY the primary outcome is psychological distress (rather than one or more of the secondary outcomes), either here or earlier in the Introduction. p. 8 line 18 who determines that a mental health problem “is likely to be a problem for them again following their release”, and how? p. 9 line 6 a web-based randomisation system is used to “ensure concealment” but if it is 1:1 potential participants may detect a pattern. Has block randomisation been considered? If not, why not? p. 10 line 16 “Outcome measure data will be collected at baseline” – this seems contradictory. The CORE-OM is the primary outcome measure – what evidence exists for its reliability and validity? The entire trial hinges on this measure, so this is critical. p. 13 line 3 small typo: “follow-ups interviews” (remove ‘s’) Table 2 lists some measures not identified as primary or secondary outcome measures (e.g., Trauma questionnaire). The primary and secondary outcome measures are also listed on page 10. Then the survey measures are listed again on page 13. Removal of significant redundancy/duplication such as this would substantially shorten the paper, without loss of information. Health economic work appears to rely on self-reported health service use. It would be useful to cite recent evidence suggests that this can be valid in ex-prisoners: Carroll et al (2016). Health & Justice, 4:11; DOI: 10.1186/s40352-016-0042-x. p. 14 line 48 assuming lifetime maintenance of short-term health gains seems incautious. Is this justifiable? Consider a Figure depicting the study design. Is the primary analysis powered to accommodate inclusion of baseline measures as covariates a priori? Is the trial powered for the secondary outcomes?
--	---

	Have the researchers considered the risk of contamination, and strategies to minimise this? If not, why not? ISRCTN registration is noted in the Abstract but not the body of the manuscript. This information should be included in the body of the manuscript.
--	--

REVIEWER	Leah Hamilton Temple University Philadelphia PA, USA
REVIEW RETURNED	19-Jul-2017

GENERAL COMMENTS	Overall the authors present a detailed protocol of an interesting and important study on offender mental health in the transition back into the community. Most of the "N/A" and "No" responses in the tick-box section of this review are due to the fact that this is a protocol not a completed study. However a few issues remain that ought to be addressed.  1. There are a few large scale/multi-site studies (although not necessarily randomized controlled trials) from America that ought to be reviewed and incorporated into the literature review that have addressed prisoner reentry. While not mental health specific, all 4 of these studies have collected data on mental health conditions of offenders, and to some extent they have addressed service coordination for offenders- including mental health substance abuse services. Please review and consider incorporating literature from: (1) the Reentry Partnership Initiative Project (e.g. Taxman, Young & Byrne, 2002); (2) the extensive publications on the SVORI Project (Serious and Violent Offenders Reentry Initiative) particularly those focussing on mental health/substance use outcomes (e.g. Lattimore & Visher, 2009; Hamilton & Belenko, 2015; Wallace et al., 2016); (3) The Returning Home Study (Mallik-Kane & Visher, 2008); and the CJ-DATS II Project (e.g. Toi & Morgo-Wilson, 2015). 2. Selection bias issues: The protocol should acknowledge and discuss the potential selection bias of 1. excluding the serious/enduring mental health problem population and 2. having only those 'willing to engage with treatment services and research procedures'. Of course, this is not to suggest that either individuals with serious mental health conditions need to be included or that participation should be mandatory. Rather it needs to be clearer why the eligibility criteria is what it is, how this may influence the outcomes, and whether anything can be done to statistically mitigate this bias. 3. Is there any justification for the intervention time-frame? It does appear to align with the RPI project, but there's no explanation given in text. 4. Are there arguments for the use of the various outcome measure instruments? For example, why is the Leeds Dependence Questionnaire used? There are a number of substance use disorder screeners available (e.g. the DAST-10) including those specifically designed for offender populations (TCU Drug Screen). Please justify the screener choice.
---

	5. For the process evaluation, will any standardized measures be developed to measure implementation fidelity? Some basic questionnaires could be used to supplement the semi-structured interviews 6. Although it is difficult to develop an in-depth statistical analysis plan prior to data collection, the multi-institution nature of the study suggests that higher level effects should be considered in the analyses. Consider multi-level modelling using nesting within region, institution or practitioners.
--	---

VERSION 1 – AUTHOR RESPONSE

Reviewer 1

Comment: In my view the manuscript could be shortened considerably, without loss of information. A careful proofread for clarity of expression would also strengthen the manuscript.

Response - we have attempted to shorten the manuscript, but due the the additional infomation requested it has actually increased in size

Comment: I would also encourage the authors to consider replacing the term 'offender' with something less pejorative. Since the population of interest is adult males released from prison, not necessarily on parole, one might argue that use of the term 'offender' in this context is somewhere between pejorative and inaccurate.

Response - The term offender has been replaced with a more appropriate term where applicable

More specific comments follow.

Comment:P. 5 line 5 "rates of 50-90%" – presumably prevalence rates? Is this point, period or lifetime prevalence?

Response - We have clarified in the text that this is point prevalence

Comment: p. 5 line 5 "trials are limited" – it might be appropriate to cite here a recent systematic review documenting precisely this: Kouyoumdjian et al (2015). Am J Public Health, 105(4), e13-e33.

Response - reference has been added and text slightly amended

Comment: P. 5 line 11 need a reference to support the statement about PTSD.

Response - statement about PTSD has been removed as it is covered later in paragraph

Comment: p. 5 line 13 note also recent evidence of high rates of self-harm after release from prison: Borschmann et al (2017). ANZ J Psychiatr, 51(3), 250-259.

Response - reference has been added

Comment: p. 5 line 16 clarify whether comorbidity refers to multiple mental illnesses, or co-occurrence of mental illness and substance use disorder, or both/either.

Response - clarified in text as comorbidity was especially evident with substance abuse

Comment: p. 5 line 21 “in our previous study” – more information is required and could be provided in a few words – for example was this a cross-sectional study of ### prisoners in the UK?

Response - additional information added regarding cross-sectional sample and sample size

Comment: p. 5 line 25 – is the SPCR study based in the UK?

Response - clarified in text that this study was based in England and Wales

Comment: p. 6 line 15 note that there is also some international evidence regarding engagement with mental health services for ex-prisoners: see for example Thomas et al (2016). Psychological Medicine, 46(3), 611-621.

Response - more detail added in text regarding international evidence

Comment: p. 6 line 49 it would be helpful to briefly explain WHY the primary outcome is psychological distress (rather than one or more of the secondary outcomes), either here or earlier in the Introduction.

Response - Further details about why the CORE-OM was chosen as the primary outcome measure (it was chosen following a consensus meeting that followed pilot testing, is quick to use and easy to understand, and maps on to the aim of the intervention). This has been added in the text with further information in a footnote

Comment: p. 8 line 18 who determines that a mental health problem “is likely to be a problem for them again following their release”, and how?

Response - a footnote has been added to provide further information regarding this matter

Comment: p. 9 line 6 a web-based randomisation system is used to “ensure concealment” but if it is 1:1 potential participants may detect a pattern. Has block randomisation been considered? If not, why not?

Response - This term has been removed, but concealment was from the research team to ensure that the researchers doing baseline assessment and randomising participants could not influence who got the intervention and who did not. Potential participants would not be able to detect any patterns as they were not aware of what previous participants were randomised to.

Comment: p. 10 line 16 “Outcome measure data will be collected at baseline” – this seems contradictory.

Response - text has been altered to remove this contradiction

Comment: The CORE-OM is the primary outcome measure – what evidence exists for its reliability and validity? The entire trial hinges on this measure, so this is critical.

Response - we have stated that the CORE-OM is a reliable and well-validated measure and provided the reference for the evidence for this.

Comment: p. 13 line 3 small typo: “follow-ups interviews” (remove ‘s’)

Response - typo has been corrected

Comment: Table 2 lists some measures not identified as primary or secondary outcome measures (e.g., Trauma questionnaire). The primary and secondary outcome measures are also listed on page 10. Then the survey measures are listed again on page 13. Removal of significant redundancy/duplication such as this would substantially shorten the paper, without loss of information.

Response - Table 2 includes some measures that are not primary and secondary outcomes, but these measures are included to describe the characteristics of the sample with plans to include them within a moderator analysis. In our view, the lists offer different information and whilst we could remove some items from the list on Page 13 to remove some duplication, some additional explanatory text would be required and this would negate any savings in the length of the manuscript. As such, our preference would be to leave the two lists as they are.

Comment: Health economic work appears to rely on self-reported health service use. It would be useful to cite recent evidence suggests that this can be valid in ex-prisoners: Carroll et al (2016). Health & Justice, 4:11; DOI: 10.1186/s40352-016-0042-x.

Response - this reference has been included along with a brief line of additional text

Comment: p. 14 line 48 assuming lifetime maintenance of short-term health gains seems incautious. Is this justifiable?

Response - We have changed text in this line to:

Comment: Improvements in lifetime gains will be linked with short-term gains seen in the trial and will be associated with social inclusion outcomes such as effective resettlement, increased employment, or reduced re-offending rates. Sustained improvements in these will be modelled based on evidence from the literature. We will test the impact of differing the duration of the persistence of any short-term gains on QALYs (and associated costs) if there is no evidence in the literature.

Comment: Consider a Figure depicting the study design.

Response - We feel the study design is already fairly clear in the text and Table 2 and, being mindful of the size of the manuscript, we have opted to not to include a figure

Comment: Is the primary analysis powered to accommodate inclusion of baseline measures as covariates a priori? Is the trial powered for the secondary outcomes?

Response - We have added the following line to the manuscript to add clarification on this point:

The sample size is based on the ability to detect a difference on the primary outcome only and not on the inclusion of baseline measures as covariate.

Comment: Have the researchers considered the risk of contamination, and strategies to minimise this? If not, why not?

Response - A paragraph has been added to deal with the issue of contamination – unfortunately this adds to the length of the manuscript.

Comment: ISRCTN registration is noted in the Abstract but not the body of the manuscript. This information should be included in the body of the manuscript.

Response - This has now been added to the body of the manuscript

Reviewer: 2

Reviewer Name: Leah Hamilton

Institution and Country: Temple University, Philadelphia PA, USA

Please state any competing interests or state 'None declared': None declared

Please leave your comments for the authors below

Overall the authors present a detailed protocol of an interesting and important study on offender mental health in the transition back into the community. Most of the "N/A" and "No" responses in the tick-box section of this review are due to the fact that this is a protocol not a completed study. However a few issues remain that ought to be addressed.

Comment: 1. There are a few large scale/multi-site studies (although not necessarily randomized controlled trials) from America that ought to be reviewed and incorporated into the literature review that have addressed prisoner reentry. While not mental health specific, all 4 of these studies have collected data on mental health conditions of offenders, and to some extent they have addressed service coordination for offenders- including mental health substance abuse services. Please review and consider incorporating literature from: (1) the Reentry Partnership Initiative Project (e.g. Taxman, Young & Byrne, 2002); (2) the extensive publications on the SVORI Project (Serious and Violent Offenders Reentry Initiative) particularly those focussing on mental health/substance use outcomes (e.g. Lattimore & Visher, 2009; Hamilton & Belenko, 2015; Wallace et al., 2016); (3) The Returning Home Study (Mallik-Kane & Visher, 2008); and the CJ-DATS II Project (e.g. Toi & Morgo-Wilson, 2015).

Response - these have been added with one additional sentence.

Comment: 2. Selection bias issues: The protocol should acknowledge and discuss the potential selection bias of 1. excluding the serious/enduring mental health problem population and 2. having only those 'willing to engage with treatment services and research procedures'. Of course, this is not to suggest that either individuals with serious mental health conditions need to be included or that participation should be mandatory. Rather it needs to be clearer why the eligibility criteria is what it is, how this may influence the outcomes, and whether anything can be done to statistically mitigate this bias.

Response - The intervention was developed specifically for prisoners with common mental health problem. Treatment options were available for prisoners with serious and enduring mental health problems (as highlighted in the introduction section) and our aim was to provide an intervention for those that had no (or inadequate) treatment available.

Having only those willing to engage with treatment services and research procedures is a basic necessity in order to undertake a trial of this nature. This is a new intervention and not available outside of the research project, so participants need to be willing to engage (at least at the outset) to agree to participate. We feel that our inclusion criteria is broad and we have tried to include all men with mental health problems who do not have established treatment pathways, including those with complex needs. We recognise that having an acceptable intervention that people want to engage with is important and as such we will be recording and reporting on all those who disengage from the intervention (and research) and we will also report on the number of people who do not want to take part in the study.

In our experience during our pilot work, reasons for not wanting to take part included those who felt they did not need the intervention because they already had a support network, as well as those who didn't want to engage. Therefore, reasons for not taking part in the study will also be recorded and reported.

Comment: 3. Is there any justification for the intervention time-frame? It does appear to align with the RPI project, but there's no explanation given in text.

Response - A sentence has been added to justify the intervention time-frame

Comment: 4. Are there arguments for the use of the various outcome measure instruments? For example, why is the Leeds Dependence Questionnaire used? There are a number of substance use disorder screeners available (e.g. the DAST-10) including those specifically designed for offender populations (TCU Drug Screen). Please justify the screener choice.

Response - The selection of primary and secondary outcome measures followed two separate consultation exercises and a period of field testing of a number of measures. This work will be described in a separate paper (yet to be submitted). A footnote stating this has been added.

Comment: 5. For the process evaluation, will any standardized measures be developed to measure implementation fidelity? Some basic questionnaires could be used to supplement the semi-structured interviews

Response - We have added a line in text stating that an intervention components checklist will be used to measure intervention fidelity

Comment: 6. Although it is difficult to develop an in-depth statistical analysis plan prior to data collection, the multi-institution nature of the study suggests that higher level effects should be considered in the analyses. Consider multi-level modelling using nesting within region, institution or practitioners.

Response - The analysis plan states that we will adjust for stratification variables. This includes site (which is equivalent to region and institution and also the practitioners are they work as teams within each region. We have added clarification within the text stating this means site